DATA RELEASE

# Chromosomal-level genome assembly of golden birdwing *Troides aeacus* (Felder & Felder, 1860)

Hong Kong Biodiversity Genomics Consortium*,†

## ABSTRACT

The golden birdwing *Troides aeacus* (Lepidoptera, Papilionidae), a significant species in Asia, faces habitat loss due to urbanization and human activities, necessitating its protection. However, the lack of genomic resources hinders our understanding of their biology and diversity, and impedes our conservation efforts based on genetic information or markers. Here, we present the first chromosomal-level genome assembly of *T. aeacus* using PacBio SMRT and Omni-C scaffolding technologies. The assembled genome (351 Mb) contains 98.94% of the sequences anchored to 30 pseudo-molecules. The genome assembly has high sequence continuity with contig length N50 = 11.67 Mb and L50 = 14, and scaffold length N50 = 12.2 Mb and L50 = 13. A total of 24,946 protein-coding genes were predicted, with high BUSCO score completeness (98.8% and 94.7% of genome and proteome BUSCO, respectively. This genome offers a significant resource for understanding the swallowtail butterfly biology and carrying out its conservation.

**Submitted:** 11 January 2024

* Correspondence on behalf of the consortium. E-mail: jeromehui@cuhk.edu.hk

† Collaborative Authors: Entomological experts who validated the dataset and their affiliations appears at the end of the document

Preprint submitted at https://doi.org/10.1101/2024.01.13.575334

Included in the series: *Hong Kong Biodiversity Genomics* (https://doi.org/10.46471/GIGABYTE_SERIES_0006)

**Subjects** Genetics and Genomics, Animal Genetics, Biodiversity

## INTRODUCTION

The golden birdwing butterfly *Troides aeacus* (Figure 1A) is a swallowtail butterfly that is widely distributed in Asia, including Bangladesh, Myanmar, Cambodia, China, India, Laos, Malaysia, Nepal, Thailand, and Vietnam [1]. The species is generally large, with a wingspan reaching ~15 cm, and has iconic black forewings and golden-yellow hindwings carved with grey stripes and black spots [2, 3]. Due to its attractive appearance, it has been vastly collected and traded in curio markets [2, 4, 5].

Similar to other homometabolans, *T. aeacus* has larvae and pupae stages: five larval instar stages before transforming into its green-girdled pupal stage [3]. The larvae are generally dependent on Aristolochiaceae host plants, especially of the genus *Aristolochia*, which can be commonly found in Asia [1–3, 6]. After emergence, the adults feed and live around nectaring flowers such as those in the genus *Hibiscus*, *Ixora*, *Lantana*, *Mussaenda*, and *Spathodea* [1, 7]. Anthropogenic activities, including deforestation, grazing, herbicide application, hunting, land reclamation, mine exploitation, and trading, have been suggested to pose threats to *T. aeacus* [1, 3, 8]. In certain places, such as Hong Kong, *T. aeacus* has also been suggested for protection and restoration efforts to recover its lost habitat. In Taiwan, the trade of endemic subspecies such as *T. aeacus* is protected by the Convention on International Trade in Endangered Species of Wild Fauna and Flora [9].

**Figure 1.** Genomic information of *Troides aeacus*.
(A) Photo of *T. aeacus*; (B) Statistics of the assembled genome; (C) Omni-C contact map of the assembly visualised using Juicebox (v1.11.08); (D) Genomescope report with k-mer = 21; (E) Repetitive elements distribution in the assembled genome.

## CONTEXT

To date, the genomic resources in the genus *Troides* are confined to *T. helena* [10] and *T. oblongomaculatus* [11]. In light of the high conservation value of *T. aeacus* and its phylogenetic importance for understanding the diversification of butterflies [12], this species has been selected for genome sequencing by the Hong Kong Biodiversity Genomics Consortium (also known as EarthBioGenome Project Hong Kong), which is formed by investigators from eight publicly funded universities in Hong Kong. Here, we report a chromosomal-level genome assembly of the golden birdwing *T. aeacus*.

## METHODS

### Sample collection and species identification

A pupa of the golden birdwing *T. aeacus* was obtained at Lui Kung Tin, Yuen Long District, Hong Kong (22.425886 °N, 114.10538 °E) in August 2022. The pupa was snap-frozen in liquid nitrogen upon collection. The frozen pupa was then ground into a fine powder and stored at −80 °C until DNA isolation. A portion of the powder was used for species molecular identification with QIAamp DNA Mini Kit (Qiagen Cat. 51306), following the provided protocol. The DNA was then used as a template for conventional PCR with the following protocol: an initial denaturation step at 95 °C for 3 minutes followed by 36 amplification cycles for denaturation of 30 seconds each at 95 °C; 30 seconds for primer annealing at 55 °C and 1 minute for extension at 72 °C; finally, an extension step at 72 °C for 3 minutes. The



reaction mixture included PCR buffer, DNA template, 2 mM dNTP, 1.5 mM $MgCl_2$, 0.4 mM of each forward and reverse primers (LCO1490: 5′-GGTCAACAAATCATAAAGATATTGG-3′, HCO2198: 5′-TAAACTTCAGGGTGACCAAAAAATCA-3′) [13], and *Taq* DNA polymerase. The PCR was performed on a T100™ thermal cycler (Bio-Rad, USA). The unpurified PCR products were sent to BG Hong Kong for Sanger sequencing. The returned sequence was validated with the chromatogram, and the resultant sequence was searched against Genbank for species validation using the BLASTN algorithm (RRID:SCR_001598).

## Isolation of high molecular weight genomic DNA

High molecular weight genomic DNA was isolated from the remaining stored powder using the Qiagen MagAttract HMW kit (Qiagen Cat. No. 67563) following the manufacturer's protocol. In summary, 1 g of sample powder was placed in a microcentrifuge tube with 200 µl 1× phosphate-buffered saline (PBS), RNase A, Proteinase K, and Buffer AL. The mixture was incubated at room temperature (22–25 °C) for 2.5 hours until the tissue was completely disintegrated. The sample was then eluted with 120 µl of elution buffer (PacBio Ref. No. 101-633-500) and stored at 4 °C. In order to keep the integrity of the DNA, wide-bore pipette tips were used for any DNA transfer during the process. The sample was then subjected to quality control with the Qubit® Fluorometer, Qubit™ dsDNA HS, and BR Assay Kits (Invitrogen™ Cat. No. Q32851). An overnight pulse-field gel electrophoresis was performed to estimate the size of the isolated DNA using three markers (λ-Hind III digest; Takara Cat. No. 3403, DL15,000 DNA Marker; Takara Cat. No. 3582A and CHEF DNA Size Standard-8-48 kb Ladder; Cat. No. 170-3707). Additionally, the sample purity was examined by the NanoDrop™ One/OneC Microvolume UV–Vis Spectrophotometer (with A260/A280: ~1.8 and A260/A230: >2.0 as a standard threshold).

## DNA shearing, PacBio library preparation, and sequencing

A total of 120 µl of DNA, corresponding to 10 µg DNA, was transferred to a g-tube (Covaris Part No. 520079). The tube was then subjected to six centrifugation steps with 2,000 × g of 2 minutes each. The resultant DNA was saved in a 2 mL DNA LoBind® Tube (Eppendorf Cat. No. 022431048) at 4 °C until library preparation. The molecular weight of the isolated DNA was examined by overnight pulse-field gel electrophoresis. The electrophoresis profile was set as follows: 5 K as the lower end and 100 K as the higher end for the designated molecular weight; Gradient = 6.0 V/cm; Run time = 15 h:16 min; included angle = 120°; Int. Sw. Tm = 22 s; Fin. Sw. Tm = 0.53 s; Ramping factor: a = Linear. The gel was run in 1.0% PFC agarose in 0.5× TBE buffer at 14 °C.

A SMRTbell library was made using the SMRTbell® prep kit 3.0 (PacBio Ref. No. 102-141-700), following the provided protocol. In summary, single-stranded overhangs of the genomic DNA were removed, and the DNA was repaired from any physical damage caused by shearing. Subsequently, both DNA ends were tailed with an A-overhang, and ligation of T-overhang SMRTbell adapters was performed at 20 °C for 30 minutes. The SMRTbell library was then purified with SMRTbell® cleanup beads (PacBio Ref. No. 102158-300). The size and concentration of the library were assessed with the pulse-field gel electrophoresis and the Qubit® Fluorometer, Qubit™ dsDNA HS, and BR Assay Kits (Invitrogen™ Cat. No. Q32851), respectively. A subsequent nuclease treatment step was carried out to remove non-SMRTbell structures in the library. A final size-selection step was performed to remove small DNA fragments in the library with 35% AMPure PB beads. The

**Table 1.** Summary of the genome sequencing data.

| Library | Reads | Bases | Coverage (X) | Accession |
|---|---|---|---|---|
| PacBio HiFi | 2,805,656 | 27,181,071,888 | 78 | SRR24631717 |
| Omni-C | 144,777,842 | 21,716,676,300 | 62 | SRR26815782 |

Sequel® II binding kit 3.2 (PacBio Ref. No. 102-194-100) was used for final preparation. In short, Sequel II primer 3.2 and Sequel II DNA polymerase 2.2 were annealed and bound to the SMRTbell library, respectively. Next, the library was loaded at an on-plate concentration of 50–90 pM using the diffusion loading mode. The sequencing was conducted on the Sequel IIe System with an internal control provided in the kit. The sequencing was performed in 30-hour movies, with 120 min pre-extension, connected to the software SMRT Link v11.0 (PacBio). HiFi reads were generated and collected for further analysis. One SMRT cell was used for this sequencing (Table 1).

## Omni-C library preparation and sequencing

An Omni-C library was made using the Dovetail® Omni-C® Library Preparation Kit (Dovetail Cat. No. 21005) according to the provided protocol. In summary, 80 mg of frozen, powered tissue sample was placed in a microcentrifuge tube with 1 mL 1× PBS and formaldehyde. The fixed DNA was digested with endonuclease DNase I. Next, the concentration and size of the digested sample were examined by the Qubit® Fluorometer, Qubit™ dsDNA HS, and BR Assay Kits (Invitrogen™ Cat. No. Q32851) and the TapeStation D5000 HS ScreenTape, respectively. Both DNA ends were polished, and ligation of the biotinylated bridge adaptor was conducted at 22 °C for 30 minutes. The subsequent proximity ligation between crosslinked DNA was performed at 22 °C for 1 hour. After ligation, the DNA was reverse crosslinked and purified with SPRIselect™ Beads (Beckman Coulter Product No. B23317) to remove the biotin that was not internal to the ligated fragments. The Dovetail™ Library Module for Illumina (Dovetail Cat. No. 21004) was used for end repair and adapter ligation. The DNA was tailed with an A-overhang, which allowed Illumina-compatible adapters to ligate to the DNA fragments at 20 °C for 15 minutes. The Omni-C library was then sheared into fragments with USER Enzyme Mix and purified with SPRIselect™ Beads. The isolation of DNA fragments with internal biotin was performed with Streptavidin Beads. Universal and Index PCR Primers from the Dovetail™ Primer Set for Illumina (Dovetail Cat. No. 25005) were used to amplify the constructed library. Size selection was carried out with SPRIselect™ Beads targeting fragments ranging between 350 bp and 1,000 bp. Finally, the concentration and fragment size of the sequencing library were examined with the Qubit® Fluorometer, Qubit™ dsDNA HS, and BR Assay Kits, and the TapeStation D5000 HS ScreenTape, respectively. The resultant library was sequenced on the Illumina HiSeq-PE150 platform (Table 1).

## Genome assembly and gene model prediction

*De novo* genome assembly was performed using Hifiasm (RRID:SCR_021069) [14]. Haplotypic duplications were identified and removed using purge_dups (RRID:SCR_021173) based on the depth of the HiFi reads [15]. Proximity ligation data from the Omni-C library were used to scaffold the genome assembly by YaHS [16]. Transposable elements (TEs) were annotated as previously described [17] using the automated Earl Grey TE annotation pipeline (version 1.2, https://github.com/TobyBaril/EarlGrey). A total of 38,780

**Table 2.** Details of the genome assembly statistics.

| Species name | *Troides aeacus* |
| --- | --- |
| total_length | 350,661,970 |
| number | 36 |
| mean_length | 9,740,610 |
| longest | 14,808,706 |
| shortest | 35,011 |
| N_count | 0.0002% |
| Gaps | 4 |
| N50 | 12,212,588 |
| N50n | 13 |
| N70 | 11,032,717 |
| N70n | 19 |
| N90 | 8,896,329 |
| N90n | 26 |
| BUSCOs (Genome, lepidoptera_odb10) | C:98.8%[S:98.6%,D:0.2%],F:0.1%,M:1.1%,n:5286 |
| HiFi (X) | 78 |
| HiFi Reads | 2,805,656 |
| HiFi Bases | 27,181,071,888 |
| HiFi Q30% | 2 |
| HiFi Q20% | 4 |
| HiFi GC% | 38 |
| HiFi Ave_len | 9,688 |
| Gene models | 23,068 |
| number of protein-coding genes | 24,946 |
| BUSCOs (Proteome, lepidoptera_odb10) | C:94.7%[S:83.9%,D:10.8%],F:0.4%,M:4.9%,n:5286 |
| total_length of protein-coding genes | 9,533,860 |
| mean_length of protein-coding genes | 382 |

papilionidae reference protein sequences were downloaded from NCBI as protein hits to perform genome annotation using Braker (v3.0.8; RRID:SCR_018964) [18] with default parameters.

## DATA VALIDATION AND QUALITY CONTROL

During DNA extraction and PacBio library preparation, the samples were subjected to quality control with NanoDrop™ One/OneC Microvolume UV–Vis Spectrophotometer, Qubit® Fluorometer, and overnight pulse-field gel electrophoresis. The Omni-C library was inspected by Qubit® Fluorometer and TapeStation D5000 HS ScreenTape.

Regarding the genome assembly, the Hifiasm output was blast to the NT database, and the resultant output was used as input for Blobtools (v1.1.1; RRID:SCR_017618) [19]. Scaffolds that were identified as possible contamination were removed from the assembly manually (Figure 2). A statistical kmer-based approach was applied to estimate the heterozygosity of the assembled genome heterozygosity. The repeat content and the corresponding sizes were analysed using Jellyfish (RRID:SCR_005491) [20] and GenomeScope (RRID:SCR_017014) [21] (Figure 1D; Table 4). Furthermore, telomeric repeats were inspected by FindTelomeres [22]. BUSCO (v5.5.0) [23] was used to assess the completeness of the genome assembly and gene annotation with the metazoan dataset (lepidoptera_odb10).

Omni-C reads and PacBio HiFi reads were used to measure the assembly completeness and consensus quality (QV) using Merqury (v1.3; RRID:SCR_022964) [24] with kmer 19, resulting in 83.111% kmer completeness for the Omni-C data and 68.2844 QV scores for the HiFi reads, corresponding to 99.9999% accuracy. Oxford synteny plots for comparison

blobtools_pacbio.blobDB.json.bestsum.phylum.p8.span.100.blobplot.bam0



**Figure 2.** Genome assembly quality control and contaminant detection.

**Table 3.** Information of 30 chromosomal-length scaffolds.

| Chromosome no. | Scaffold name | Scaffold length (bp) | Sum of the percentage of the whole genome |
|---|---|---|---|
| 1 | scaffold_1 | 14,808,706 | 4.22% |
| 2 | scaffold_2 | 14,665,000 | 8.41% |
| 3 | scaffold_3 | 14,313,961 | 12.49% |
| 4 | scaffold_4 | 14,238,555 | 16.55% |
| 5 | scaffold_5 | 14,017,748 | 20.55% |
| 6 | scaffold_6 | 13,917,518 | 24.51% |
| 7 | scaffold_7 | 13,815,727 | 28.45% |
| 8 | scaffold_8 | 13,654,990 | 32.35% |
| 9 | scaffold_9 | 13,495,232 | 36.20% |
| 10 | scaffold_10 | 13,141,228 | 39.94% |
| 11 | scaffold_11 | 12,752,029 | 43.58% |
| 12 | scaffold_12 | 12,274,249 | 47.08% |
| 13 | scaffold_13 | 12,212,588 | 50.56% |
| 14 | scaffold_14 | 12,154,636 | 54.03% |
| 15 | scaffold_15 | 12,125,450 | 57.49% |
| 16 | scaffold_16 | 11,669,766 | 60.82% |
| 17 | scaffold_17 | 11,660,701 | 64.14% |
| 18 | scaffold_18 | 11,537,742 | 67.43% |
| 19 | scaffold_19 | 11,032,717 | 70.58% |
| 20 | scaffold_20 | 10,726,358 | 73.64% |
| 21 | scaffold_21 | 10,597,782 | 76.66% |
| 22 | scaffold_22 | 10,099,347 | 79.54% |
| 23 | scaffold_23 | 9,998,358 | 82.39% |
| 24 | scaffold_24 | 9,129,920 | 84.99% |
| 25 | scaffold_25 | 9,045,561 | 87.57% |
| 26 | scaffold_26 | 8,896,329 | 90.11% |
| 27 | scaffold_27 | 8,628,000 | 92.57% |
| 28 | scaffold_28 | 7,826,594 | 94.80% |
| 29 | scaffold_29 | 7,424,256 | 96.92% |
| 30 | scaffold_30 | 7,076,422 | 98.94% |

**Table 4.** GenomeScope result summary (k-mer = 21).

| Property | Min | Max |
|---|---|---|
| Homozygous (aa) | 97.04% | 97.11% |
| Heterozygous (ab) | 2.89% | 2.96% |
| Genome Haploid Length (bp) | 265,767,007 | 268,320,884 |
| Genome Repeat Length (bp) | 69,335,920 | 70,002,201 |
| Genome Unique Length (bp) | 196,431,086 | 198,318,683 |
| Model Fit | 78.57% | 99.17% |
| Read Error Rate | 0.73% | 0.73% |

to the same genus genome *T. helena* (GCA_029286815.1) and *T. oblongomaculatus* (GCA_029032895.1) were generated using the R package 'ggplot2' [25] as described in Lee *et al.* [26, 27] (Figure 3, 4 and Table 7).

## RESULTS AND DISCUSSION

### Genome assembly of T. aeacus

A total of 27 Gb of HiFi bases were yielded with an average HiFi read length of 9,688 bp with 78X coverage (Supplementary Information 1). After incorporating 21.7 Gb Omni-C data, the resulting genome assembly was 350.66 Mb in size with 36 scaffolds, 30 of which are of chromosome length (Figure 1B–C; Table 2, 3). The genome has high contiguity with a

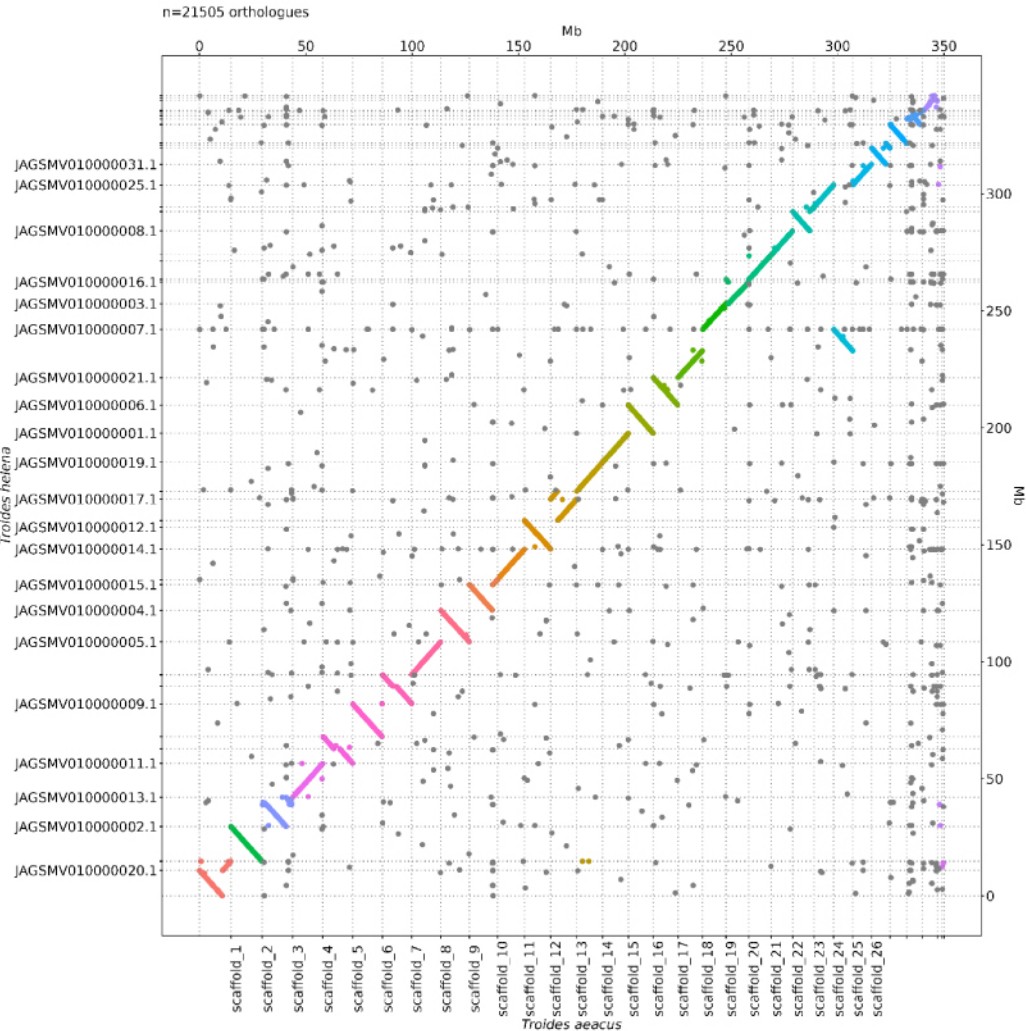

**Figure 3.** Oxford synteny plots with *Troides Helena*.

scaffold N50 value of 12.21 Mb, and high completeness with a complete BUSCO (RRID:SCR_015008) estimation of 98.8% (lepidoptera_odb10) (Figure 1B, Table 2). While the genome size estimation was about 268.3 Mb with a 2.93% nucleotide heterozygosity rate (Figure 1D; Table 4), the assembled *T. aeacus* genome has a genome size similar to other swallowtail butterfly genomes, including *T. helena* (~330 Mb) [10] and *T. oblongomaculatus* (~348 Mb) [11]. In addition, 43 telomeres were found in 25 scaffolds of the assembly genome (Table 5). Furthermore, 23,068 gene models were predicted with a BUSCO score of 94.7% (lepidoptera_odb10).

## Repeat content

A total repetitive content of 29.50% was identified in the assembled genome, including 5.16% unclassified elements (Figure 1E; Table 6). Among the known repeats, long interspersed nuclear elements (LINEs) were the most abundant ones (12.01%), followed by short interspersed nuclear element (SINE) retrotransposons (6.38%) and DNA transposons

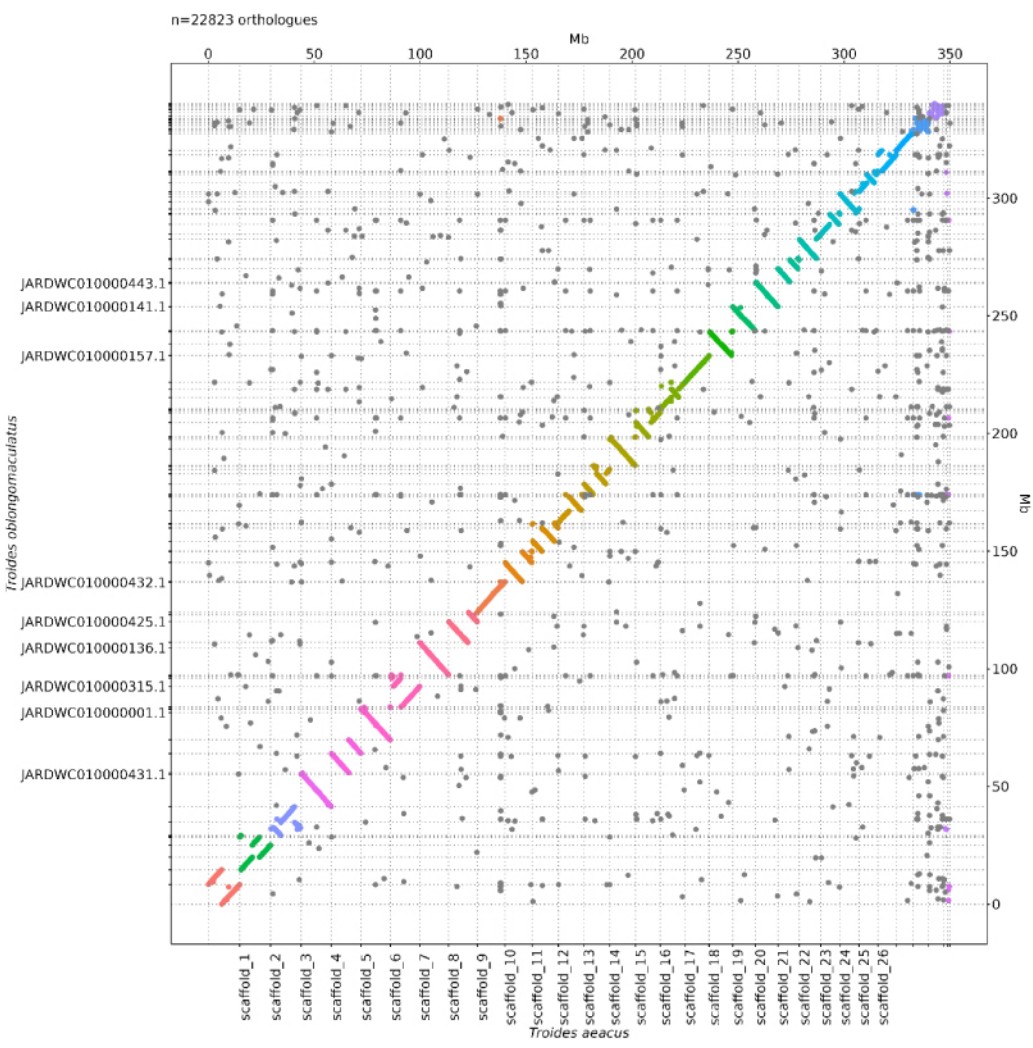

**Figure 4.** Oxford synteny plots with _Troides oblongomaculatus_.

(4.71%). In contrast, Rolling Circle, long terminal repeats (LTRs), Penelope, and others were present in low proportions (Rolling Circle: 0.78%, LTR: 0.26%, Penelope: 0.20%, other: 0.02%).

## CONCLUSION AND REUSE POTENTIAL

This study presents the first chromosomal-level genome assembly of the golden birdwing _T. aeacus_, a useful and precious resource for further phylogenomic studies of birdwing butterfly species in terms of species diversification and conservation.

## DATA AVAILABILITY

The final genome assembly was submitted to NCBI under the accession number (GCA_033220335.2). The raw reads yielded from this study were deposited on the NCBI database under the BioProject accession number PRJNA973839. The genome annotation files were deposited in figshare [28].

**Table 5.** Summary of telomeric repeats found in 25 scaffolds.

| Scaffold name | Strand | Position at scaffold | Sequence |
|---|---|---|---|
| scaffold_1 | Forward | start | taacctaacctaacctaacctaacctaacctaacctaacctaacctaacc |
| scaffold_2 | Reverse | end | aggttagtaggtaggttaggtaggttagttaggtaggttaggttaggtta |
| scaffold_4 | Forward | start | aacctaacctaacctaacctaacctaacctaacctaacctaacctaacct |
| scaffold_4 | Reverse | end | aggttaggtaggttaggttagtttaggttaggttaggttaggttaggtta |
| scaffold_5 | Forward | start | acctaacctaacctaacctaacctaacctaacctaacctaacctaaccta |
| scaffold_5 | Reverse | end | gttaggttaggttaggttaggttaggttaggttaggttaggttaggttag |
| scaffold_6 | Forward | start | taacctaacctaacctaacctaacctaacctaacctaacctaacctaacc |
| scaffold_6 | Reverse | end | aggttaggtaggttaggttaggttaggttaggttaggttaggttaggtt |
| scaffold_7 | Forward | start | acctaacctaacctaacctaacctaacctaacctaacctaacctaaccta |
| scaffold_7 | Reverse | end | taggttaggttaggttaggttaggttaggttaggttaggttaggttaggt |
| scaffold_8 | Forward | start | taacctaacctaacctaacctaacctaacctaacctaacctaacctaacc |
| scaffold_8 | Reverse | end | aggttaggttaggttaggttaggttaggttaggttaggttaggttaggtt |
| scaffold_9 | Forward | start | ctaacctaacctaacctaacctaacctaacctaacctaacctaacctaac |
| scaffold_9 | Reverse | end | taggttaggttaggttaggttaggttaggttaggttaggttaggttaggt |
| scaffold_10 | Forward | start | aacctaacctaacctaacctaacctaacctaacctaacctaacctaacct |
| scaffold_10 | Reverse | end | ttaggttaggttaggttaggttaggttaggttaggttaggttaggttagg |
| scaffold_11 | Forward | start | cctaacctaacctaacctaacctaacctaacctaacctaacctaacctaa |
| scaffold_12 | Forward | start | aacctaacctaacctaacctaacctaacctaacctaacctaacctaacct |
| scaffold_12 | Reverse | end | aggttaggttaggttaggttaggttaggttaggttaggttaggttaggtt |
| scaffold_14 | Forward | start | aacctaacctaacctaacctaacctaacctaacctaacctaacctaacct |
| scaffold_14 | Reverse | end | aggttaggttaggttaggttaggttaggttaggttaggttaggttaggtt |
| scaffold_15 | Forward | start | taacctaacctaacctaacctaacctaacctaacctaacctaacctaacc |
| scaffold_16 | Reverse | end | ggttaggttaggttaggttaggttaggttaggttaggttaggttaggtta |
| scaffold_17 | Forward | start | aacctaacctaacctaacctaacctaacctaacctaacctaacctaacct |
| scaffold_17 | Reverse | end | ggttaggttaggttaggttaggttaggttaggttaggttaggttaggtta |
| scaffold_18 | Forward | start | acctaacctaacctaacctaacctaacctaacctaacctaacctaaccta |
| scaffold_18 | Reverse | end | ggttaggttaggttaggttaggttaggttaggttaggttaggttaggtta |
| scaffold_19 | Forward | start | taacctaacctaacctaacctaacctaacctaacctaacctaacctaacc |
| scaffold_19 | Reverse | end | aggttaggttaggttaggttaggttaggttaggttaggttaggttaggtt |
| scaffold_21 | Forward | start | cctaacctaacctaacctaacctaacctaacctaacctaacctaacctaa |
| scaffold_21 | Reverse | end | ttaggttaggttaggttaggttaggttaggttaggttaggttaggttagg |
| scaffold_22 | Forward | start | taacctaacctaacctaacctaacctaacctaacctaacctaacctaacc |
| scaffold_22 | Reverse | end | taggttaggttaggttaggttaggttaggttaggttaggttaggttaggt |
| scaffold_23 | Forward | start | acctaacctaacctaacctaacctaacctaacctaacctaacctaaccta |
| scaffold_23 | Reverse | end | aggttaggttaggttaggttaggttaggttaggttaggttaggttaggtt |
| scaffold_24 | Forward | start | cctaacctaacctaacctaacctaacctaacctaacctaacctaaccaac |
| scaffold_24 | Reverse | end | taggttaggttaggttaggttaggttaggttaggttaggttcggttaggt |
| scaffold_25 | Forward | start | ctaacctaacctaacctaacctaacctaacctaacctaacctaacctaac |
| scaffold_25 | Reverse | end | taggtaggtaggtaggtaggtaggtaggtaggtaggtaggtaggtaggta |
| scaffold_26 | Reverse | end | ggttaggttaggttaggttaggttaggttaggttaggttaggttaggtta |
| scaffold_27 | Reverse | end | taggttaggttaggttaggttaggttaggttaggttaggttaggttaggt |
| scaffold_28 | Forward | start | taacctaacctaacctaacctaacctaacctaacctaacctaacctaacc |
| scaffold_28 | Reverse | end | ttaggttaggttaggttaggttaggttaggttaggttaggttaggttagg |

## ABBREVIATIONS

PBS, phosphate-buffered saline; LINE: long interspersed nuclear element; LTR: long terminal repeat; QV: consensus quality; SINE: short interspersed nuclear element; TE: transposable element.

## DECLARATIONS

## Ethics approval and consent to participate

The authors declare that ethical approval was not required for this type of research.

**Table 6.** Summary of the repetitive elements in the genome.

| Classification | Total length (bp) | Count | Proportion (%) | No. of distinct classifications |
|---|---|---|---|---|
| DNA | 16,507,235 | 16,573 | 4.71 | 2,128 |
| LINE | 42,109,363 | 120,899 | 12.01 | 6,176 |
| LTR | 906,819 | 1,286 | 0.26 | 771 |
| Other (Simple Repeat, Microsatellite, RNA) | 55,850 | 178 | 0.02 | 86 |
| Penelope | 692,645 | 363 | 0.20 | 119 |
| Rolling Circle | 2,727,115 | 12,594 | 0.78 | 614 |
| SINE | 22,365,985 | 86,794 | 6.38 | 1,216 |
| Unclassified | 18,097,624 | 35,455 | 5.16 | 3,824 |
| **SUM** | **103,462,636** | **274,142** | **29.50** | **14,934** |

**Table 7.** Statistics of *Troides* genomes.

| Species name | *Troides aeacus* (Pacbio_only_version) | *Troides aeacus* | *Troides helena* | *Troides oblongomaculatus* |
|---|---|---|---|---|
| total_length | 350,661,170 | 350,661,970 | 346,252,535 | 343,353,597 |
| number | 37 | 36 | 284 | 457 |
| mean_length | 9,477,329 | 9,740,610 | 1,219,199 | 751,321 |
| longest | 25,391,358 | 14,808,706 | 20,604,617 | 13,649,974 |
| shortest | 35,011 | 35,011 | 4,987 | 526 |
| N_count | 0 | 800 | 0 | 4,538 |
| Gaps | 0 | 4 | 0 | 46 |
| N50 | 12,154,636 | 12,212,588 | 11,016,300 | 5,909,187 |
| N50n | 12 | 13 | 13 | 20 |
| N70 | 11,032,717 | 11,032,717 | 9,113,214 | 4,315,177 |
| N70n | 18 | 19 | 20 | 33 |
| N90 | 7,826,594 | 8,896,329 | 4,177,244 | 1,457,309 |
| N90n | 26 | 26 | 30 | 55 |
| metazoa_odb10 | / | C:98.9%[S:98.3%, D:0.6%], F:0.2%, M:0.9%, n:954 | C:96.6%[S:96.0%, D:0.6%], F:0.4%, M:3.0%, n:954 | C:98.8%[S:98.2%, D:0.6%], F:0.3%, M:0.9%, n:954 |
| insecta_odb10 | / | C:98.9%[S:98.5%, D:0.4%], F:0.4%, M:0.7%, n:1367 | C:96.7%[S:96.5%, D:0.2%], F:0.4%, M:2.9%, n:1367 | C:99.1%[S:98.7%, D:0.4%], F:0.4%, M:0.5%, n:1367 |
| lepidoptera_odb10 | / | C:98.8%[S:98.6%, D:0.2%], F:0.1%, M:1.1%, n:5286 | C:96.4%[S:96.2%, D:0.2%], F:0.3%, M:3.3%, n:5286 | C:98.8%[S:98.6%, D:0.2%], F:0.2%, M:1.0%, n:5286 |
| **Species** | ***Troides aeacus*** | ***Troides helena*** | ***Troides oblongomaculatus*** | |
| Number of Proteins | 24,946 | 24,366 | 23,414 | |
| Sum of Amino Acids | 9,533,860 | 9,754,783 | 9,382,566 | |
| Mean of Proteins | 382 | 400 | 401 | |
| Sum of Exons(bp) | 28,601,578 | 29,264,202 | 28,147,634 | |
| Mean of Exons | 220 | 223 | 218 | |
| Sum of Introns(bp) | 92,410,884 | 96,334,286 | 93,619,605 | |
| Mean of Introns | 880 | 901 | 884 | |
| Numer of gene loci | 23,068 | 22,338 | 21,500 | |
| Sum of gene region (bp) | 105,638,137 | 107,935,950 | 105,567,879 | |
| % of gene loci in genome | 30.13% | 31.17% | 30.75% | |
| Average gene region(bp) | 4,579 | 4,832 | 4,910 | |
| metazoa_odb10 | C:90.5%[S:84.6%, D:5.9%], F:1.0%, M:8.5%, n:954 | C:86.2%[S:80.4%, D:5.8%], F:1.4%, M:12.4%, n:954 | C:90.4%[S:84.0%, D:6.4%], F:1.3%, M:8.3%, n:954 | |
| insecta_odb10 | C:93.3%[S:82.7%, D:10.6%], F:0.4%, M:6.3%, n:1367 | C:89.0%[S:78.0%, D:11.0%], F:1.1%, M:9.9%, n:1367 | C:93.4%[S:82.7%, D:10.7%], F:0.6%, M:6.0%, n:1367 | |
| lepidoptera_odb10 | C:94.7%[S:83.9%, D:10.8%], F:0.4%, M:4.9%, n:5286 | C:90.9%[S:79.6%, D:11.3%], F:0.8%, M:8.3%, n:5286 | C:94.9%[S:83.8%, D:11.1%], F:0.5%, M:4.6%, n:5286 | |

## Competing interests

The authors declare that they do not have competing interests.

## Authors' contributions

JHLH, TFC, LLC, SGC, CCC, JKHF, JDG, SCKL, YHS, CKCW, KYLY, and YW conceived and supervised the study. WLS carried out the DNA extraction, library preparation, and sequencing. WN performed genome assembly and gene model prediction. HSFP, WKY, CYLC, SSSC, KKLM, and HYY collected and maintained the butterfly samples. All authors approved the final version of the manuscript.

## Funding

This work was funded and supported by the Hong Kong Research Grant Council Collaborative Research Fund (C4015-20EF), CUHK Strategic Seed Funding for Collaborative Research Scheme (3133356), and CUHK Group Research Scheme (3110154).

## Acknowledgements

The authors thank Sean Law for the discussions.

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

## DETAILS OF COLLABORATIVE AUTHORS

### • List of authors in Hong Kong Biodiversity Genomics Consortium

Jerome H. L. Hui,[1] Ting Fung Chan,[2] Leo Lai Chan,[3] Siu Gin Cheung,[4] Chi Chiu Cheang,[5,6] James Kar-Hei Fang,[7] Juan Diego Gaitan-Espitia,[8] Stanley Chun Kwan Lau,[9] Yik Hei Sung,[10,11] Chris Kong Chu Wong,[12] Kevin Yuk-Lap Yip,[13,14] Yingying Wei,[15] Wai Lok So,[1] Wenyan Nong,[1] Hydrogen Sui Fai Pun,[16] Wing Kwong Yau,[16] Colleen Yuk Lin Chiu,[16] Sammi Shan Shan Chan,[16] Kacy Ka Ling Man,[16] Ho Yin Yip[1]

[1]School of Life Sciences, Simon F.S. Li Marine Science Laboratory, State Key Laboratory of Agrobiotechnology, Institute of Environment, Energy and Sustainability, The Chinese University of Hong Kong, Hong Kong, China
[2]School of Life Sciences, State Key Laboratory of Agrobiotechnology, The Chinese University of Hong Kong, Hong Kong SAR, China
[3]State Key Laboratory of Marine Pollution and Department of Biomedical Sciences, City University of Hong Kong, Hong Kong SAR, China



[4]State Key Laboratory of Marine Pollution and Department of Chemistry, City University of Hong Kong, Hong Kong SAR, China

[5]Department of Science and Environmental Studies, The Education University of Hong Kong, Hong Kong SAR, China

[6]EcoEdu PEI, Charlottetown, PE, C1A 4B7, Canada

[7]Department of Food Science and Nutrition, Research Institute for Future Food, and State Key Laboratory of Marine Pollution, The Hong Kong Polytechnic University, Hong Kong SAR, China

[8]The Swire Institute of Marine Science and School of Biological Sciences, The University of Hong Kong, Hong Kong SAR, China

[9]Department of Ocean Science, The Hong Kong University of Science and Technology, Hong Kong SAR, China

[10]Science Unit, Lingnan University, Hong Kong SAR, China

[11]School of Allied Health Sciences, University of Suffolk, Ipswich, IP4 1QJ, UK

[12]Croucher Institute for Environmental Sciences, and Department of Biology, Hong Kong Baptist University, Hong Kong SAR, China

[13]Department of Computer Science and Engineering, The Chinese University of Hong Kong, Hong Kong SAR, China

[14]Sanford Burnham Prebys Medical Discovery Institute, La Jolla, CA, USA

[15]Department of Statistics, The Chinese University of Hong Kong, Hong Kong SAR, China

[16]Fung Yuen Butterfly Reserve, Hong Kong SAR, China

