## [Reviewer Report]

Comments on revised manuscriptThe paper has substantially been enhanced after the first revision. I suggest that this manuscript can be published after the following minor revisions: 1.L279: ‘formosanus’ is also part of the scientific name which should be Italic type. 2.It’s recommended to beautify the figures and tables.

---

## [Editor Report]

Editor’s AssessmentThis work is part of a series of papers from the Hong Kong Biodiversity Genomics Consortium sequencing the rich biodiversity of species in Hong Kong. This example presents the genome of the golden birdwing butterfly Troides aeacus (Lepidoptera, Papilionidae). A notable and popular species in Asia that faces habitat loss due to urbanization and human activities. The lack of genomic resources impedes conservation efforts based on genetic markers, as well as better understanding of its biology. Using PacBio HiFi long reads and Omni-C a 351Mb genome was assembled genome anchored to 30 pseudo-molecules. After reviewers requested more information on the genome quality it seems there was high sequence continuity with contig length N50 = 11.67 Mb and L50 = 14, and scaffold length N50 = 12.2 Mb and L50 = 13. Allowing a total of 24,946 protein-coding genes were predicted. This study presents the first chromosomal-level genome assembly of the golden birdwing T. aeacus, a potentially useful resource for further phylogenomic studies of birdwing butterfly species in terms of species diversification and conservation.

---

## [Reviewer Report]

Reviewer name and names of any other individual's who aided in reviewer Dr. Kumar Saurabh SinghDo you understand and agree to our policy of having open and named reviews, and having your review included with the published papers. (If no, please inform the editor that you cannot review this manuscript.)YesIs the language of sufficient quality?YesPlease add additional comments on language quality to clarify if needed
Are all data available and do they match the descriptions in the paper? YesAdditional CommentsAre the data and metadata consistent with relevant minimum information or reporting standards? See GigaDB checklists for examples <a href="http://gigadb.org/site/guide" target="_blank">http://gigadb.org/site/guide</a>NoAdditional Comments1. I've noticed that the genome assembly file has been uploaded to NCBI, but I couldn't locate the corresponding annotation files in GFF format. Additionally, I couldn't find gene models for Troides aeacus on NCBI or any other platform. As per Giga Science data policy, these files should be made publicly available.  2. The paper lacks information on the contig N50 and L50, although I did find this data on NCBI. Is there a specific reason for omitting the contig N50/L50 details from the main text or tables?Is the data acquisition clear, complete and methodologically sound?YesAdditional CommentsIs there sufficient detail in the methods and data-processing steps to allow reproduction?YesAdditional Comments1. I have noticed that the QV value is missing for the given assembly. To assess the base-level accuracy of your assembly, the authors should calculate the consensus quality (QV), comparing the frequency of k-mers present in the raw Omni-C reads (as you only have short-reads from Omni-c) with those present across the final assembly perhaps using Merqury. 2. Incorporating Omni-c data did not result in a significant increase in the contig N50. Have you identified any specific reasons for this outcome? 3. The overall BUSCO completeness for proteins appears to be disproportionately low (~86%) compared to genomic completeness (~98%). Could this be attributed to the absence of RNAseq data for predicting accurate gene models?  
Is there sufficient data validation and statistical analyses of data quality? YesAdditional CommentsI believe it's essential to assess the assembly quality through comparative genomic analyses, a component seemingly missing from the manuscript. While the text mentions the availability of genomic resources within the same genus, conducting a genome-wide comparison of these assemblies could provide valuable insights into the overall synteny and contiguity of the T. aeacus assembly. To ensure annotation consistency, it's important to compare genome assemblies by generating distributions of intron/exon lengths for annotations across multiple assemblies.Is the validation suitable for this type of data?YesAdditional CommentsIs there sufficient information for others to reuse this dataset or integrate it with other data?YesAdditional CommentsAny Additional Overall Comments to the AuthorRecommendationMinor Revision

---

## [Reviewer Report]

Upload additional filesDRR-202401-01/form/GigabyteDRR-202401-01-comments.docxReviewer name and names of any other individual's who aided in reviewer Xueyan LiDo you understand and agree to our policy of having open and named reviews, and having your review included with the published papers. (If no, please inform the editor that you cannot review this manuscript.)YesIs the language of sufficient quality?YesPlease add additional comments on language quality to clarify if needed
Are all data available and do they match the descriptions in the paper? YesAdditional CommentsAre the data and metadata consistent with relevant minimum information or reporting standards? See GigaDB checklists for examples <a href="http://gigadb.org/site/guide" target="_blank">http://gigadb.org/site/guide</a>YesAdditional CommentsIs the data acquisition clear, complete and methodologically sound?YesAdditional CommentsIs there sufficient detail in the methods and data-processing steps to allow reproduction?NoAdditional CommentsIs there sufficient data validation and statistical analyses of data quality? YesAdditional CommentsIs the validation suitable for this type of data?YesAdditional CommentsIs there sufficient information for others to reuse this dataset or integrate it with other data?YesAdditional CommentsAny Additional Overall Comments to the AuthorRecommendationMajor Revision